# Hepatoprotective and Neuroprotective Effects of Naringenin against Lead-Induced Oxidative Stress, Inflammation, and Apoptosis in Rats

**DOI:** 10.3390/biomedicines11041080

**Published:** 2023-04-03

**Authors:** Lubna A. H. Mansour, Gehad E. Elshopakey, Fatma M. Abdelhamid, Talat A. Albukhari, Samah J. Almehmadi, Bassem Refaat, Mohamed El-Boshy, Engy F. Risha

**Affiliations:** 1Department of Clinical Pathology, Faculty of Veterinary Medicine, Mansoura University, Mansoura 35516, Egypt; 2Department of Immunology and Hematology, Faculty of Medicine, Umm Al-Qura University, Makkah P.O. Box 6165, Saudi Arabia; 3Laboratory Medicine Department, Faculty of Applied Medical Sciences, Umm Al-Qura University, Al Abdeyah, Makkah P.O. Box 7607, Saudi Arabia

**Keywords:** naringenin, lead acetate, neurotoxicity, hepatotoxicity, apoptosis, oxidative stress

## Abstract

Naringenin (NRG) is one of the most important naturally occurring flavonoids, predominantly found in some edible fruits, such as citrus species and tomatoes. It has several biological activities, such as antioxidant, antitumor, antiviral, antibacterial, anti-inflammatory, antiadipogenic, and cardioprotective effects. The heavy metal lead is toxic and triggers oxidative stress, which causes toxicity in many organs, including the liver and brain. This study explored the potential protective role of NRG in hepato- and neurotoxicity caused by lead acetate in rats. Four groups of ten male albino rats were included: group 1 was a control, group 2 was orally treated with lead acetate (LA) at a dose of 500 mg/kg BW, group 3 was treated with naringenin (NRG) at a dose of 50 mg/kg BW, and group 4 was treated with 500 mg/kg LA and 50 mg/kg NRG for 4 weeks. Then, blood was taken, the rats were euthanized, and liver and brain tissues were collected. The findings revealed that LA exposure induced hepatotoxicity with a significant increase in liver function markers (*p* < 0.05). In addition, albumin and total protein (TP) and the albumin/globulin ratio (A/G ratio) (*p* < 0.05) were markedly lowered, whereas the serum globulin level (*p* > 0.05) was unaltered. LA also induced oxidative damage, demonstrated by a significant increase in malonaldehyde (MDA) (*p* < 0.05), together with a pronounced antioxidant system reduction (SOD, CAT, and GSH) (*p* < 0.05) in both liver and brain tissues. Inflammation of the liver and brain caused by LA was indicated by increased levels of nuclear factor kappa beta (NF-κβ) and caspase-3, (*p* < 0.05), and the levels of B-cell lymphocyte-2 (BCL-2) and interleukin-10 (IL-10) (*p* < 0.05) were decreased. Brain tissue damage induced by LA toxicity was demonstrated by the downregulation of the neurotransmitters norepinephrine (NE), dopamine (DA), serotonin (5-HT), and creatine kinase (CK-BB) (*p* < 0.05). Additionally, the liver and brain of LA-treated rats displayed notable histopathological damage. In conclusion, NRG has potential hepato- and neuroprotective effects against lead acetate toxicity. However, additional research is needed in order to propose naringenin as a potential protective agent against renal and cardiac toxicity mediated by lead acetate.

## 1. Introduction

Lead is a poisonous metal that is used in numerous sectors. Humans are among those exposed to lead through the ingestion or inhalation of polluted food, water, soil, air, and other substances [1].

In addition to the central nervous system, which is one of the most seriously impacted organs, lead diffuses into soft tissues and harms the kidneys and liver, as well as the hematological, reproductive, and cardiovascular systems [2]. An imbalance between the production and elimination of ROS (reactive oxygen species) in tissue and cellular components is the process underlying lead-induced oxidative stress, which damages membranes, DNA, and proteins [3]. The second way in which lead causes oxidative stress in cells is its impact on their antioxidant defense mechanisms. Through the inhibition of functional SH groups in various enzymes, including δ-aminolevulinic acid dehydratase (ALAD), superoxide dismutase (SOD), catalase (CAT), glutathione peroxidase (GPx), and glucose-6-phosphate dehydrogenase (G6PD), lead has been found to change antioxidant activity [4]. One of its negative consequences is the excessive generation of reactive oxygen and nitrogen species [5,6]. Lead is recognized as an immunotoxic suppressor that interacts, changes function, accumulates, and damages the immune response by inducing cell death [7]. Brain damage demonstrated in the cerebellum, cerebral cortex, and hippocampus is associated with lead intoxication through the production of excess free radicals that induce alterations in neurotransmitters [8]. Low levels of lead exposure have been linked to abnormal behavior, diminished hearing, the degeneration of the neuromuscular system, learning impairments, and poor cognitive ability in both experimental animals and people [2]. Additionally, lead may also cause an increase in NF-κβ and IL-10 production, which in turn causes further neurotoxicity [9]. Lead toxicity is a type of toxicity that causes an increase in the vital components of apoptosis, caspase-3 and Bcl2 [10]. Lead toxicity alters the normal morphology of hepatic lobules, cord-like structures of living liver cells, and results in sinusoidal congestion and hyperchromatic hepatocytes [11]. It has also been linked to a significant increase in total bilirubin, the generation of free radicals, and the action of liver enzymes [12].

Flavonoids, which are a subclass of naturally occurring polyphenolic compounds, can be identified by their flavan nucleus. These types of chemicals are most frequently present in fruits, vegetables, and plant-based beverages. Flavonoids in dietary supplements are believed to boost health and prevent disease. Current nutraceutical, pharmaceutical, medical, cosmetic, and other products all make extensive use of them.

One of the most significant natural flavonoids is naringenin (NRG) [13]. NRG is the aglycone form of naringin (4′,5,7-trihydroxyflavanone-7-rhamnoglucoside), which belongs to the flavonoid class known as flavanones and is found mainly in citrus fruits, including lemon, orange, mandarin, and grapefruit [14,15]. A promising medicinal compound, NRG’s biological effects on human health include increased antioxidant defenses, anti-inflammatory effects, the regulation of immune system activity, antiatherogenic effects, and the scavenging of reactive oxygen species. It also decreases lipid peroxidation biomarkers and protein carbonylation [16]. NRG can cross the blood–brain barrier and exert different neuronal effects through its ability to interact with the protein kinase C signaling pathway [17]. In addition, it has pharmacologically effective antioxidant, anti-inflammatory, anticancer, antimutagenic, antiatherogenic, anticarcinogenic, hepatoprotective, and neuroprotective effects [18].

Studies have also been conducted concerning the protective effect of NRG against experimental LA-induced hepatic and neurotoxicity [19]. The objective of this study was to evaluate the beneficial effect of NRG on hepatic and neurotoxicity caused by LA in rats by assessing liver function tests, brain marker levels, and antioxidant system and oxidative stress markers, as well as the expression levels of apoptotic markers in the liver and brain, together with histopathological changes.

## 2. Materials and Methods

### 2.1. Chemical Compounds

Lead acetate (CH_3_CO_2_)_2_ Pb.3H_2_O, 99.99% pure (Item No.: 316512), and carboxy methyl cellulose (CMC) 0.5% powder (Product NO, 25698) were acquired from El-Nasr Pharmaceutical Chemical Company, Cairo, Egypt. Naringenin 95% (4′, 5, 7-trihydroxyflavanone) C15H12O5 (Item No.: N5893) was purchased from Sigma-Aldrich (St. Louis, MO, USA). Additional chemicals and reagents were purchased from Diamond Diagnostics (Cairo, Egypt).

### 2.2. Experimental Planning and Animals

The Benha Farm of Laboratory Animals provided 40 healthy adult male albino rats, which were kept in a typical laboratory environment (12/12 h light/dark cycle, 25 ± 1 °C). Prior to the test, the rats were allowed two weeks of acclimation and unrestricted access to clean water and food. The Animal Research Ethical Committee of the Faculty of Veterinary Medicine at Mansoura University in Egypt developed guidelines for all animal studies, and these regulations were adhered to throughout the research process (Approval Number: M/18; April 2019).

For four weeks, rats were randomly divided into 4 separate groups:

Group 1 (the negative group), in which the animals were administered the usual water and food pellets.

Group 2 (the LA-positive group), in which the animals received a daily dose of 500 mg/kg/BW of LA dispersed in distilled water through a stomach tube [20].

Group 3 (the NRG control group), in which the animals were supplied with NRG orally dissolved in carboxy methyl cellulose (CMC) 0.5% at a dose of 50 mg/kg/BW/day for 4 weeks through a stomach tube [21].

Group 4, in which the animals were treated with LA plus NRG, using the same duration and dose as previously mentioned.

### 2.3. Collection and Processing of Samples

At the end of the experiment, blood samples were obtained from the eye canthi of the rats and placed in plain test tubes; dipotassium salt of EDTA anticoagulant was used to collect the first sample for hematological examination. Plain test tubes were loaded with additional blood samples. After being allowed to coagulate for 20 min at room temperature, the samples were stored at 4 °C for four hours. The samples were initially centrifuged at 3000× *g* for 5 min to obtain serum for the biochemical and immunological assays. The rats were euthanized by dislocating their cervical vertebrae.

One gram of both liver and brain tissues from each animal was then swiftly removed, weighed, and homogenized in ice-cold PBS after being rinsed in ice-cold saline buffer (20 mM Tris-HCl, 0.14 M NaCl buffer, pH 7.4). In order to further examine the oxidative stress marker, antioxidant properties, and gene expression in the liver and brain using RNA, the homogenate was centrifuged for 15 min at 1800× *g* at a temperature of −4 °C. The supernatants were carefully collected and stored at −20 °C. The liver and brain samples were preserved in neutral buffered formaldehyde at a 10% concentration for histological examination.

### 2.4. Biochemical Tests

#### 2.4.1. Evaluation of Liver Markers

Serum liver transaminase enzyme (ALT and AST) activities were measured using kits from Human (Magdeburg, Germany). Spinreact (Barcelona, Spain) kits were used to assess alkaline phosphatase (ALP). Direct bilirubin, albumin, total bilirubin, globulin, and total protein were measured using commercial kits obtained from Human (Magdeburg, Germany). All were determined using a spectrophotometer (BM, Magdeburg, Germany, model 5010) in accordance with the methods recommended by the manufacturer.

#### 2.4.2. Evaluation of Neural Monoamines

The modified method set out by Ciarlone (1978) uses the fluorometric technique (Jenway 6200) to assess the levels of norepinephrine (NE), serotonin (5-HT), and dopamine (DA) in brain tissue homogenate. This method is relatively fast and uses manageable sample volumes. The method has the additional advantage of saving 33% of the time, compared to the time required to accomplish a similar analysis that was reported previously. The brain tissue homogenate was examined for creatine kinase-BB utilizing Biomed kits (Cairo, Egypt).

#### 2.4.3. Evaluation of Hepatic and Neural Oxidant/Antioxidant Parameters

Using commercial kits purchased from Bio-diagnostic (Cairo, Egypt), reduced glutathione (GSH), catalase (CAT), malondialdehyde (MDA), and superoxide dismutase (SOD) activities in liver and brain tissues were evaluated spectrophotometrically in accordance with the enclosed pamphlets.

### 2.5. Analysis of Gene Expression

#### 2.5.1. Extraction of Total RNA and Reverse Transcription

Following the manufacturer’s technique, the Trizol reagent was used to extract total RNA from the liver and brain tissues (Direct-zolTM RNA MiniPrep, catalog no. R2050 Zymo, Irvine, CA, USA). The purity of RNA samples (1 µL) was verified by measuring their absorbance using a NanoDrop spectrophotometer (ND-1000, Thermo Scientific, Foster City, CA, USA) at 260 and 280 nm, and RNA ratios (A260:A280) greater than 1.8 were used for further experiments. cDNA was obtained from each sample following the manufacturer’s instructions (SensiFastTM cDNA synthesis kit, Bioline, catalog No. Bio-65053 Bioline GmbH, Luckenwalde, Germany).

The reaction mixture was composed of up to 1 μg of total RNA and contained 20 μL of DNase-free water, 4 μL of 5× Trans Amp buffer, 1 μL of reverse transcriptase, and 20 μL of DNase-free water. The final reaction mixture was then put into a thermal cycler and subjected to the following process: primer annealing at 25 °C for 10 min, reverse transcription at 42 °C for 15 min, and inactivation at 85 °C for 5 min. The samples were then stored 4 °C.

#### 2.5.2. Quantitative Real-Time PCR

Real-time PCR was used to measure the relative levels of IL-10, NF, Bcl-2, and caspase-3 mRNA (2× SensiFastTM SYBR, Bioline, catalog no. Bio-98002,Bioline GmbH, Luckenwalde, Germany). The B-Actin gene is frequently utilized as a housekeeping gene. The reaction mixture, which was performed in a total volume of 20 μL, was composed of 3 μL of cDNA, 0.8 μL of each primer, 10 μL of 2× SensiFast SYBR, 5.4 μL of d.d. water, and 10 μL of 2× SensiFast SYBR. The PCR cycling conditions for the sequences specified in Table 1 were an annealing temperature for 20 s, 95 °C for 10 min, 94 °C for 15 s, and 72 °C for 20 s. Following the amplification stage, a melting curve analysis was carried out to verify the specificity of the PCR product. Using the 2-Ct technique, the relative gene expression of B-Actin was compared to that of each sample and the control (Table 1) [22].

### 2.6. Histopathological Evaluation

The sections of liver and brain tissues that had been fixed in 10% formaldehyde were embedded in paraffin and stained with hematoxylin and eosin (H&E) [28]. An Apex biological microscope (Chippenham, Wiltshire, UK) was used to view the slides, and an Apex minigap was used to take photographs (Chippenham, Wiltshire, UK).

### 2.7. Analysis of Statistics

The data were analyzed using one-way analysis of variance (ANOVA) as mean ± standard error of the mean, then the values obtained were entered into SPSS software (New York, NY, USA, version 26), and Duncan multiple comparison tests were carried out. At *p* < 0.05, the findings were considered statistically significant.

## 3. Results

### 3.1. Protective Effect of Naringenin (NRG) on Liver Function Markers in Lead-Treated Rats

Compared with the control rats, rats that had taken LA had significantly increased serum levels of the enzymes alkaline phosphatase (ALP), aspartate aminotransferase (AST), and alanine aminotransferase (ALT). Additionally, in comparison with the control group, the total and direct bilirubin levels of the LA-intoxicated group were considerably higher. In contrast, there was minimal variation in indirect bilirubin among the groups studied. In comparison with the control group, the LA-intoxicated group showed markedly lower levels of total protein and albumin and a lower albumin/globulin ratio (A/G ratio). The serum globulin level remained unchanged. The NRG treatment significantly restored the altered serum levels of ALT, direct bilirubin, ALP, AST, albumin, and total protein in the LA + NRG-treated group to their normal levels. However, when compared with the control or LA-intoxicated groups, there was no appreciable change in indirect bilirubin (Figure 1A–F; Figure 2A–D).

### 3.2. Protective Effect of Naringenin (NRG) on the Levels of Neurotransmitters in the Brains of Lead-Intoxicated Rats

As shown in Figure 3A–D, LA-intoxicated rats exhibited lower levels of dopamine (DA), creatine kinase (CK-BB), serotonin (5-HT), and norepinephrine (NE) in comparison with the control group. This effect was reversed after the administration of NRG in the LA + NRG group compared with the untreated group. 

### 3.3. Protective Effect of Naringenin (NRG) on Hepatic and Neural Lipid Peroxidation and Antioxidant System in Lead-Intoxicated Rats

When compared with the control, LA-intoxicated rats showed a considerable increase in the levels of MDA in the liver and brain and a significant decrease in the antioxidant biomarkers (SOD, CAT, and GSH). However, NRG treatment led to a marked increase in brain GSH values, bringing them back to normal levels while leaving SOD and CAT activities unaltered in comparison with both the LA-intoxicated rats and the control (Figure 4 and Figure 5).

### 3.4. Protective Effect of Naringenin (NRG) on Hepatic and Neural Expression of Cytokines and Apoptotic Markers in Lead-Intoxicated Rats

Compared with the control, the liver and brain of LA-intoxicated rats had markedly higher levels of nuclear factor kappa beta (NF-κβ) and cysteine-aspartic proteases (caspase-3) but significantly lower levels of B-cell lymphoma 2 (BCL-2) and interleukin-10 (IL-10). Compared with the LA-intoxicated rats, NRG administration led to an improvement in the altered cytokines and apoptosis-regulatory genes, with the exception of IL-10, as shown in Figure 6 and Figure 7, respectively. 

### 3.5. Protective Effect of Naringenin (NRG) on Histopathological Alterations in Hepatic and Brain Tissues of Lead-Intoxicated Rats

#### 3.5.1. Hepatic Tissue

The hepatic histological changes and lesion scores in the tested groups are shown in Figure 8A–D. In the LA-intoxicated group, there was significant mononuclear cell infiltration in the portal area, but the LA + NRG group showed mild mononuclear cell infiltration in this area.

#### 3.5.2. Brain Tissue

When compared with the cerebral cortex sections of the control group, which displayed normal neurons, glial cells, and neuropil, some of the neurons from the LA group showed shrinkage. However, there were few neurons showing shrinkage in the cerebral cortex portions from the LA + NRG group (Figure 9A–D). Hippocampal sections showed normal neurons in the pyramidal layer (PL) in the control and NRG groups. Hippocampal sections from the LA group showed the marked degeneration and shrinkage of neurons in the pyramidal layer (PL). Hippocampal sections from the LA + NRG group showed mild degeneration and shrinkage of a few neurons in the pyramidal layer (PL) (Figure 10A–D). Cerebellar sections showed normal neurons in the granular (G), Purkinje (P), and molecular (M) cell layers in the control and NRG groups. Cerebellar sections from the LA-intoxicated group showed multiple vacuolations and the prominent loss of neurons in the Purkinje layer. Cerebellar sections from the LA + NRG group showed the mild focal loss of neurons in the Purkinje layer (Figure 11A,B1,B2,C,D).

We have summarized the hepatoprotective and neuroprotective effects of naringenin against lead acetate toxicity in rats in Figure 12.

## 4. Discussion

One of the most harmful and toxic metals, lead is linked to a variety of health issues in both people and animals, including physiological, mental, and behavioral problems [29,30]. One of the primary mechanisms causing harm to the brain, liver, kidneys, and other organs has been identified as lead-toxicity-related oxidative stress [20]. The liver, a vital organ, has a key function in the detoxification and biotransformation of biotoxins [31]. According to a 2007 examination of lead-exposed people by Mudipalli, of all the soft tissues, liver tissue has the highest lead concentration per gram of wet tissue (33%). Lead intoxication disrupts the normal architectural arrangement of the hepatic lobules, and functioning liver cells lose their characteristic cord-like structure. Hyperchromatic hepatocytes with sporadic vacuolations and sinusoidal congestion also occur [32]. Lead primarily acts in the central nervous system (CNS), and exposure to it is linked to a number of neurobehavioral and psychiatric changes [33]. According to recent studies, lead exposure has a deleterious impact on a number of neurotransmitter systems and can have a severe impact on many different fields of healthcare, particularly on the developing brain [34,35]. Naringenin (NRG) is one of the most prevalent flavonoids found in citrus fruits, tomatoes, and grapes [36]. It has a wide range of biological effects on both human and animal health, which include lowering biomarkers of lipid peroxidation and protein carbonylation, boosting antioxidant defenses, scavenging reactive oxygen species, protecting the nervous system, regulating immune system activity, and having antiatherogenic and anti-inflammatory effects [16]. Our findings demonstrated the hepato- and neuroprotective action of naringenin against lead damage.

Increased liver enzymes are markers of liver damage because they are released into the blood as a result of the loss of hepatocyte membrane integrity through lead exposure, which also results in lipid peroxidation [37,38]. In this investigation, LA-induced hepatotoxicity led to a considerable rise in the serum levels of liver enzymes (ALT, AST, and ALP) compared with the control rats. These alterations following lead exposure could be the result of cell membrane modifications triggered by oxidative stress that cause the extended release of liver enzymes into the blood [39]. Elevated ALP levels in the LA groups further indicated that subchronic lead exposure might have damaged hepatobiliary tract cells, which might in turn have resulted in bile duct blockage and infiltrative liver disease [40]. ALP has a role in several metabolic activities, including the synthesis of proteins, phospholipids, and nucleic acids; thus, the change in ALP activity may also have an impact on these processes [41]. Furthermore, our findings are consistent with those of Ref. [30], in which rats were exposed to LA for 6 weeks at a dose of 500 mg/L.

In addition, LA-intoxicated rats demonstrated a marked decline in the levels of total protein and albumin, together with a marked rise in total and direct bilirubin. Because plasma proteins, particularly albumin (Alb), are predominantly synthesized in the liver, the significant drop in serum total protein also indicated liver disease [42]. Lead disrupts a significant number of hepatocyte enzymes by attaching to plasma proteins, which prevents hepatocytes from producing proteins [43]. Additionally, it decreases the quantity of free amino acids used to make proteins and interferes with intracellular Ca^+2^ signaling, which damages the endoplasmic reticulum [44,45]. Therefore, elevated serum total bilirubin resulted from the impairment of bilirubin excretion caused by liver injury and/or hemolysis [46]. Increased hemolysis and liver damage might have contributed to the increased bilirubin levels seen in the LA-intoxicated group. Similar results were noted when rats were treated orally with LA at a dose of 60 mg/kg for 28 days [47].

This study demonstrated that NRG has hepatoprotective properties against liver damage caused by LA, with the improvement of liver function biomarkers showing that alterations in serum AST, ALP, and ALT, direct bilirubin, total protein, and albumin were restored to their normal values. This might be due to the anti-inflammatory effect of naringenin on inflammatory cytokines such as TNF-α and NF- κB [48]. These outcomes are consistent with those of Ref. [49], in which NRG was administered to rats at a dose of 50 mg/kg/BW for 4 weeks. Similarly, it has been reported that naringenin decreases liver marker enzymes during methyl nitrosamine-induced hepatotoxicity. This effect is associated with the ability of NRG to stabilize the cell membrane during hepatic damage via its anti-lipoperoxidation activity. Another possible mechanism by which NRG restores hepatocellular integrity is its ability to selectively inhibit eicosanoid synthesis, thereby reducing inflammatory responses [50]. Additionally, naringenin elicited an anti-necrotic preventive response to hepatocellular damage [51].

Based on previous analyses and the current results, lead-induced neurotoxicity might be primarily triggered by oxidative stress, which interferes with neurotransmitter release. Lead may also have an impact on serotonin (5-hydroxytryptamine; 5-HT) and other monoamine levels, which are widely dispersed throughout the brain according to some evidence [52]. In addition, our experiments demonstrated that LA-induced neurotoxicity was associated with a number of possibly harmful changes, such as lowered levels of dopamine (DA), creatine kinase (CK), norepinephrine (NE), and serotonin (5-HT). The histopathological abnormalities in the brain cells of LA-intoxicated rats provided evidence of these alterations. Our findings are partially consistent with those of Ref. [53], in which rats were treated with LA intraperitoneally for 7 days at a dose of 100 mg/kg/BW/day. Similarly, Ref. [1] reported the same results in rats exposed to LA intraperitoneally at a dose of 20 mg/kg for 7 days.

In this study, NRG repaired the aberrant expression of neurotransmitter biomarkers and restored brain tissue integrity, demonstrating its protective impact on LA-induced brain injury. Because it crossed the blood–brain barrier and had a direct protective effect on cortical cells, NRG also increased the antioxidant defense of the brain, decreased LPO levels, restored the histoarchitecture of the brain, and increased the levels of the neurotransmitter biomarkers norepinephrine (NE), dopamine (DA), and serotonin. Ref. [54] administered rats various doses (25, 50, and 100 mg/kg) of NRG for 5 days and confirmed its protective effect.

In our investigation, rats treated with LA displayed a noticeable drop in CAT and SOD activity as well as the level of GSH, together with a considerable increase in MDA levels in liver and brain homogenates. Another indication of lead-induced damage is oxidative stress, which can either increase free radical formation or deplete the antioxidant enzyme system [29,30,38]. Lead-oxidized 5-aminolevulinic acid (ALA) causes an excess of ROS (superoxide anion and hydrogen peroxide) to be produced. As a result of the inhibition of ALAD activity, these ROS accumulate in tissues [55,56]. Additionally, lead binds to other proteins containing SH groups, such as the SOD and CAT enzymes, causing a decrease in their activity and an increase in lipid peroxidation and DNA damage [29]. Lead also binds to the GSH thiol group, which is excreted in bile, in an irreversible manner [5]. SOD is the first step in cellular defense to eliminate ROS; it converts superoxide (O_2_^−^) and water (H_2_O) into hydrogen peroxide (H_2_O_2_) and molecular oxygen (O_2_). H_2_O_2_ is catalyzed by catalase into H_2_O and O_2_ [57]. The lower SOD and catalase activity that was reported in LA-exposed rats may be due to the interaction between lead and the cofactors copper, zinc, and iron. These divalent ions can be influenced by lead, which can then attach to the enzymes and impair their activities. Ref. [58] reported equivalent outcomes in the antioxidant enzyme levels of rats treated with LA at doses of 200, 300, and 400 ppm for 2 weeks.

Therefore, it is the rise in the activity of fundamental enzymes involved in the production of glutamyl cysteine, which raises the level of intracellular GSH, that causes the antioxidant activity of flavonoid compounds [59]. Naringenin has been shown to have antioxidant properties, scavenging free radicals and inhibiting nonenzymatic lipid peroxidation to prevent damage to cellular components [60,61]. Naringenin also has an antioxidant function because of its hydroxyl substituents (OH), which are highly reactive toward reactive nitrogen species (RNS) and reactive oxygen species (ROS) [62]. Therefore, the coadministration of lead with naringenin caused a significant increase in SOD, GSH, and CAT levels and a significant reduction in the level of MDA. The antioxidant effect of naringenin in rats exposed to NRG at doses of 50 and 100 mg/kg/BW orally for 20 days was previously described in [63].

The balance of anti- and proapoptotic proteins in the body can be restored by substituting lead for divalent ions (such as calcium) [64]. Lead can also induce specifically targeted apoptosis in the rat liver and brain, according to [65]. Inter-membrane proteins, such as cytochrome c, were eventually released from the mitochondria as a result of ROS, which were mostly formed in the mitochondria, attacking membrane phospholipids and causing the loss of mitochondrial membrane potential [66]. DNA splitting, nuclear chromatin condensation, and cell death were all caused by caspase-3 activation [67]. Our study revealed that, after LA administration, the proinflammatory cytokine NF-κβ and proapoptotic-related proteins such as caspase-3 were significantly upregulated in the liver and brain, whereas the proinflammatory cytokine IL-10 and proapoptotic-related proteins such as BCL-2 were downregulated in the hepatic and brain tissues. The authors of [45,68] revealed that lead treatment raises inhibitory AMPK phosphorylation and enhances NF-κβ activation, which causes excessive ROS and the upregulation of inflammatory cytokines in the blood and tissues. In a similar manner, Ref. [8] administered LA for 7 days at a dose of 20 mg kg/I/P, and this revealed significantly higher levels of the proapoptotic protein caspase-3 and the proinflammatory cytokine NF-κβ; in contrast, the proinflammatory cytokine IL-10 and the proapoptotic-related protein BCL-2 were downregulated after LA treatment. These results are completely consistent with our findings.

Our findings demonstrated that the coadministration of NRG with LA dramatically reduced the high levels of brain proinflammatory cytokines, showing that this treatment had anti-inflammatory effects. The activation of nuclear factor-κβ (NF-κβ) and the inhibition of cytokine production achieved this effect. Similarly, Ref. [63] downregulated elevated brain pro-apoptotic-related proteins such as caspase-3 [21] by preventing the production of ROS, which in turn prevented NF-κβ downstream signaling. This resulted in the modulation of the proinflammatory cytokine IL-10 and proapoptotic-related proteins such as BCL-2 [69]. Our outcomes are consistent with a previous study Ref. [70] that reported that NRG reduced brain proinflammatory cytokines (NF-κβ and IL-10) when rats were exposed to NRG at a dose of 50 mg/kg orally 1 h after the first radiation dose and then daily for 2 weeks.

## 5. Conclusions

This research suggests that the concurrent administration of NRG protects rats from LA-induced hepato- and neurotoxicity by improving liver function and reestablishing normal levels of monoamines in the brain. Additionally, NRG can act by attenuating lipid peroxidation, restoring the activities of antioxidant enzymes, and alleviating inflammation and apoptosis by modulating NF-κB, Bcl-2, and caspase-3 in both hepatic and neural tissues. However, future research is needed to fully understand the mechanistic action of NRG against lead toxicity and to determine its protective potential against renal and cardiac toxicity mediated by lead acetate.

## Figures and Tables

**Figure 1 biomedicines-11-01080-f001:**
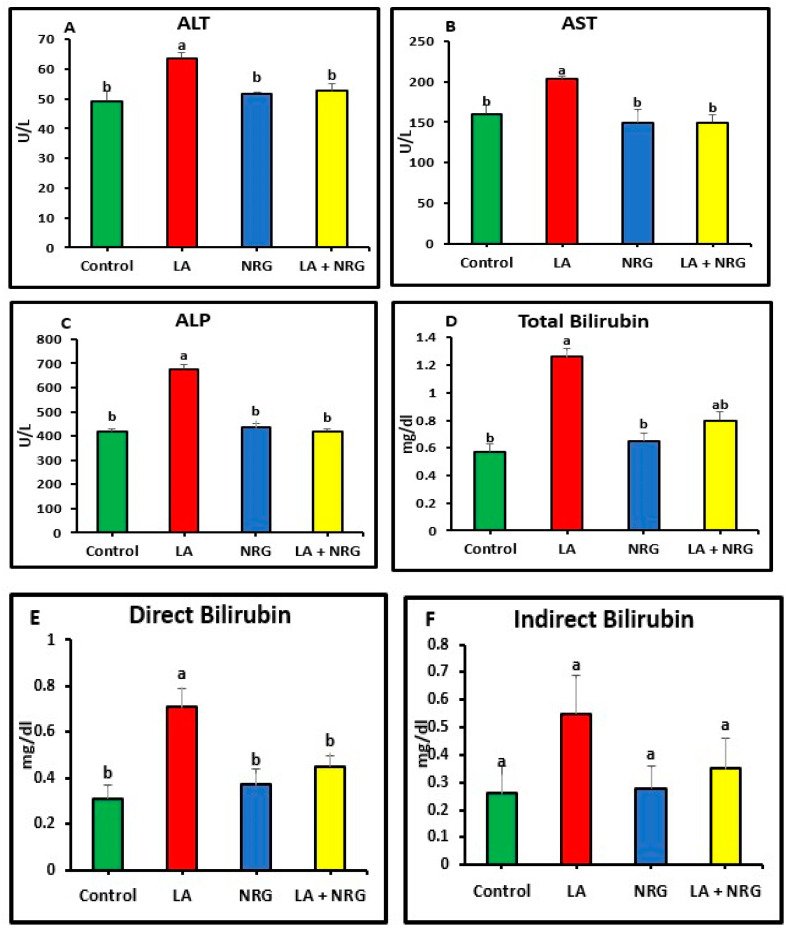
Liver function parameters: (**A**) AST (U/L), (**B**) ALT (U/L), (**C**) ALP (U/L), (**D**) total (mg/dL), (**E**) direct bilirubin (mg/dL), and (**F**) indirect bilirubin (mg/dL) at the 4th week of treatment of LA-intoxicated rats with NRG (mean ± SE)). The means in each chart with different superscripts are significantly different (*p* < 0.05). ALT: alanine aminotransferase; AST: aspartate aminotransferase; ALP: alkaline phosphatase. a for the high value; while b for the lower value.

**Figure 2 biomedicines-11-01080-f002:**
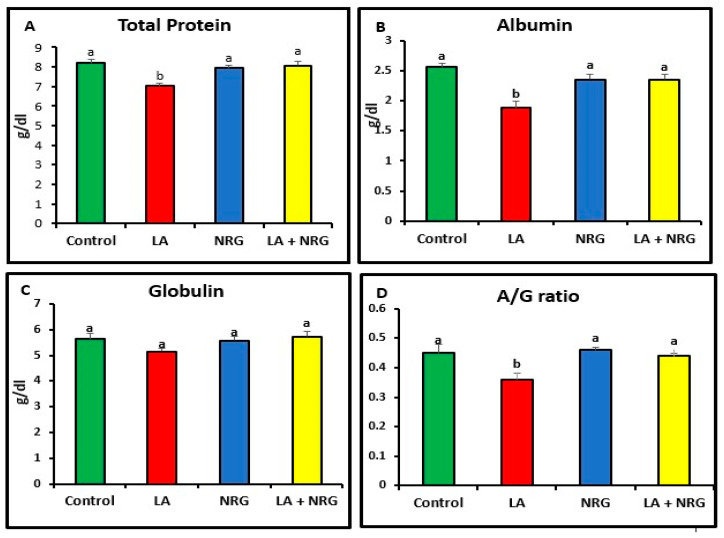
Proteinogram: (**A**) total protein (g/dL), (**B**) albumin (g/dL), (**C**) globulin (g/dL), and (**D**) A/G ratio at the 4th week of treatment of LA-intoxicated rats with NRG (mean ± SE). The means in each chart with different superscripts are significantly different (*p* < 0.05). A/G ratio: albumin/globulin ratio. a for the high value; while b for the lower value.

**Figure 3 biomedicines-11-01080-f003:**
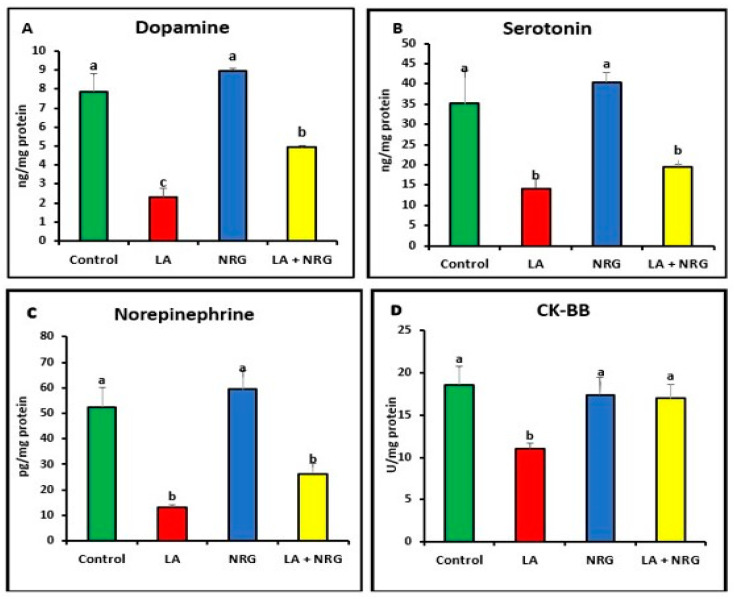
Brain neurotransmitters: (**A**) dopamine (ng/mg protein), (**B**) serotonin (ng/mg protein), (**C**) norepinephrine (pg/mg protein), and (**D**) CK-BB (U/mg/protein) in the different experimental groups at the 4th week of treatment of LA-intoxicated rats with NRG (mean ± SE). The means in each chart with different superscripts are significantly different (*p* < 0.05). CK-BB: creatine kinase brain band isoenzyme. a for the high value; b, for intermediate values between a and c; while c for the lower value.

**Figure 4 biomedicines-11-01080-f004:**
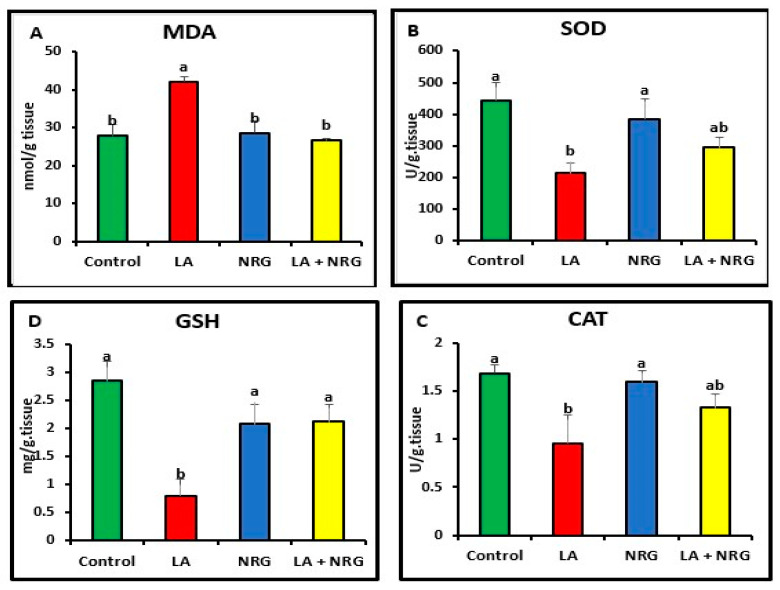
Hepatic oxidative/antioxidative parameters: (**A**) MDA (nmol/g tissue), (**B**) SOD (U/g tissue), (**C**) catalase (U/g tissue), and (**D**) GSH (mg/g tissue) at the 4th week of treatment of LA-intoxicated rats with NRG (mean ± SE)). The means in each chart with different superscripts are significantly different (*p* < 0.05). MDA: malondialdehyde; SOD: superoxide dismutase; CAT: catalase; GSH: reduced glutathione. a for the high value; while b for the lower value.

**Figure 5 biomedicines-11-01080-f005:**
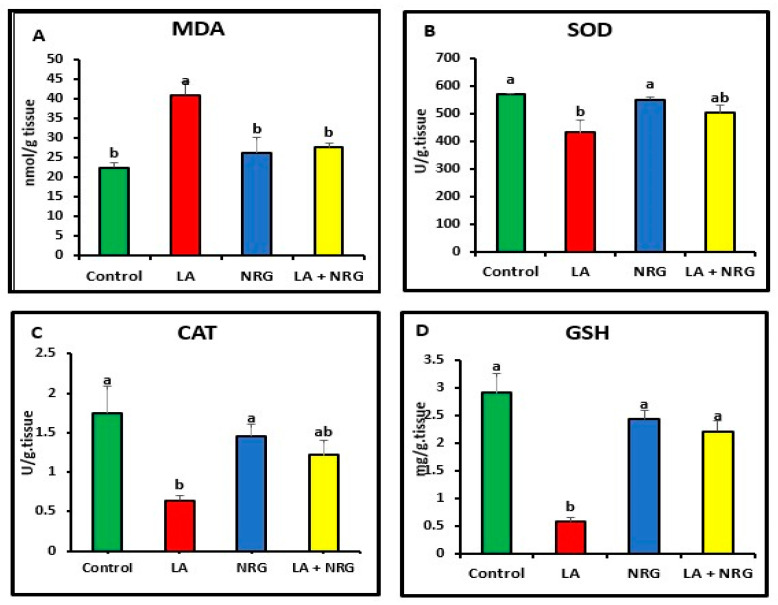
Brain oxidative/antioxidative parameters: (**A**) MDA (nmol/g tissue), (**B**) SOD (U/g tissue), (**C**) catalase (U/g tissue), and (**D**) GSH (mg/g tissue) at the 4th week of treatment of LA-intoxicated rats with NRG (mean ± SE). The means in each chart with different superscripts are significantly different (*p* < 0.05). MDA: malondialdehyde; SOD: superoxide dismutase; CAT: catalase; GSH: reduced glutathione. a for the high value; while b for the lower value.

**Figure 6 biomedicines-11-01080-f006:**
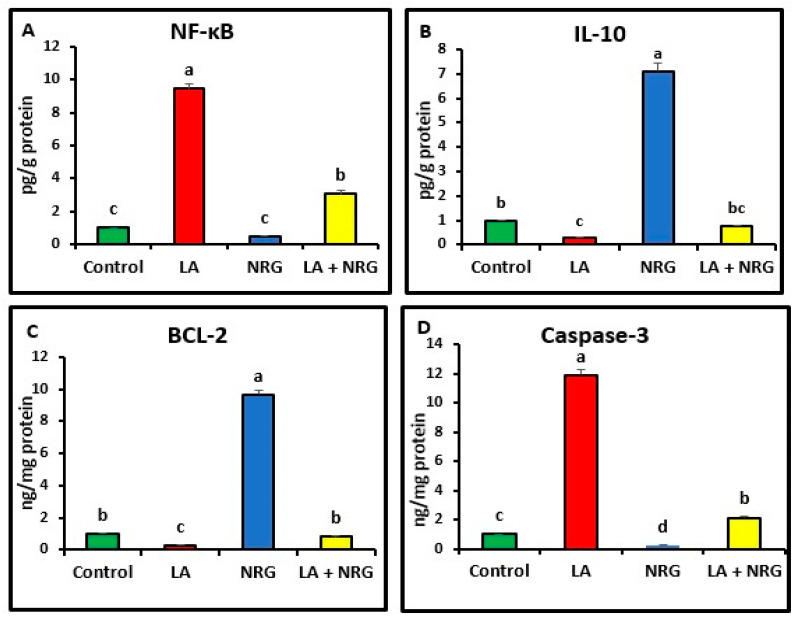
Hepatic pro- and anti-inflammatory mediators: (**A**) NF-κB (Pg/g protein), (**B**) IL-10 (Pg/g protein), (**C**) BCL-2 mRNA expression (ng/mg protein), and (**D**) caspase-3 mRNA expression (ng/mg protein) in the different experimental groups at the 4th week of treatment of LA-intoxicated rats with NRG (mean ± SE). The means in each chart with different superscripts are significantly different (*p* < 0.05). NF-κβ: nuclear factor kappa-B; IL-10: interleukin 10; BCL-2: B-cell lymphoma 2; caspase-3: cysteine-aspartic proteases. a for the high value; b, c for intermediate values between a and d, while d for the lower value.

**Figure 7 biomedicines-11-01080-f007:**
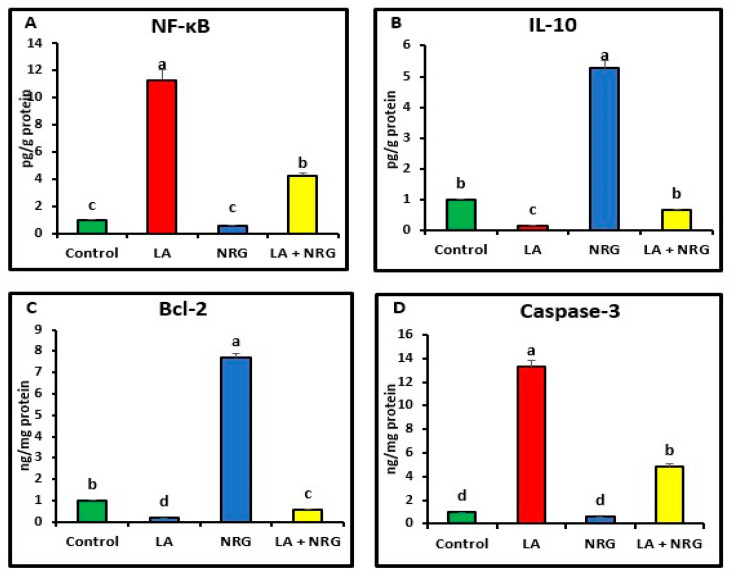
Brain pro- and anti-inflammatory mediators: (**A**) NF-κB (Pg/g protein), (**B**) IL-10 (Pg/g protein), (**C**) BCL-2, and (**D**) caspase-3 in the different experimental groups at the 4th week of treatment of LA-intoxicated rats with NRG (mean ± SE). The means in each chart with different superscripts are significantly different (*p* < 0.05). NF-κβ: nuclear factor kappa-B; IL-10: interleukin 10; BCL-2: B-cell lymphoma 2; caspase-3: cysteine-aspartic proteases. a for the high value; b, c for intermediate values between a and d, while d for the lower value.

**Figure 8 biomedicines-11-01080-f008:**
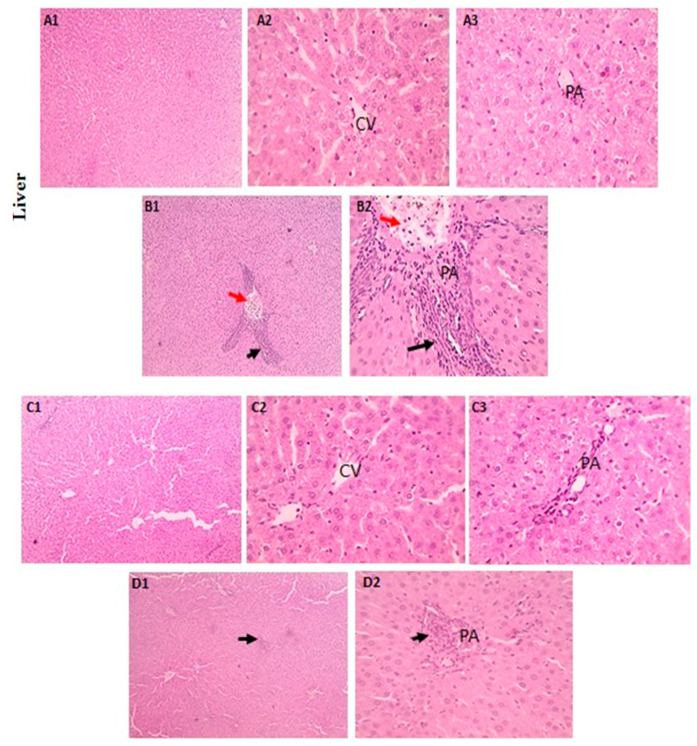
Histopathological examination of liver sections using H&E: (**A1**–**A3**) control and (**C1**–**C3**) liver sections showing the regular arrangement of hepatic cords around central veins (CVs) with normal portal areas (PAs) and sinusoids (H&E, 100× and 400×). (**B1**,**B2**) Liver sections showing portal congestion (red arrows) with massive mononuclear cell infiltration (black arrows) (H&E, 100×). (**D1**,**D2**) Liver sections showing mild mononuclear cell infiltration (black arrows) in the portal area (H&E, 400×).

**Figure 9 biomedicines-11-01080-f009:**
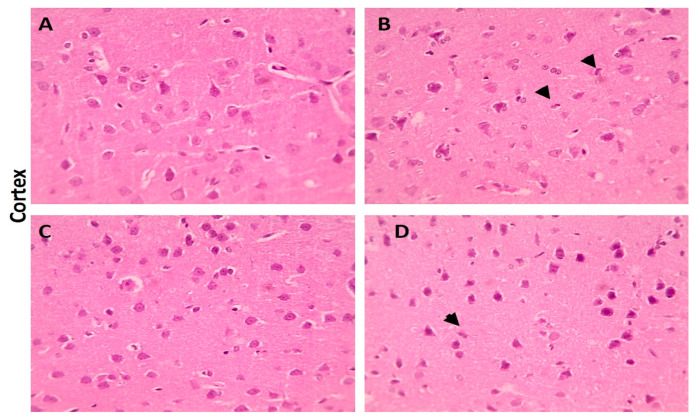
Histopathological examination of cerebral cortical sections using H&E. (**A**,**C**) Cerebral cortical section showing normal neurons, glial cells, and neuropil (H&E, 400×). (**B**) Cerebral cortical sectionshowing shrinkage of some neurons (black arrowheads) (H&E, 400×). (**D**) Cerebral cortical section showing shrinkage of a few neurons (black arrowheads) (H&E, 400×).

**Figure 10 biomedicines-11-01080-f010:**
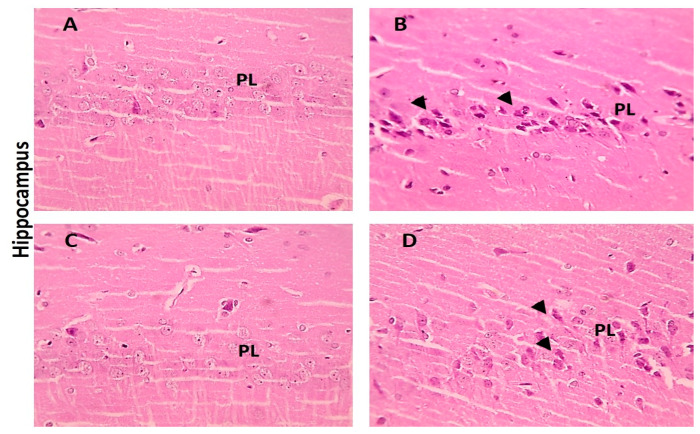
Histopathological examination of hippocampal sections using H&E. (**A**,**C**) Hippocampal section showing normal neurons in the pyramidal layer (PL) (H&E, 400×). (**B**) Hippocampal section showing marked degeneration and shrinkage of neurons in the pyramidal layer (PL) (H&E, 400×). (**D**) Hippocampal section showing mild degeneration and shrinkage of a few neurons in the pyramidal layer (PL) (black arrowheads) (H&E, 400×). (G) granular cell layer, (P) Purkinje layer and (M) molecular cell layer.

**Figure 11 biomedicines-11-01080-f011:**
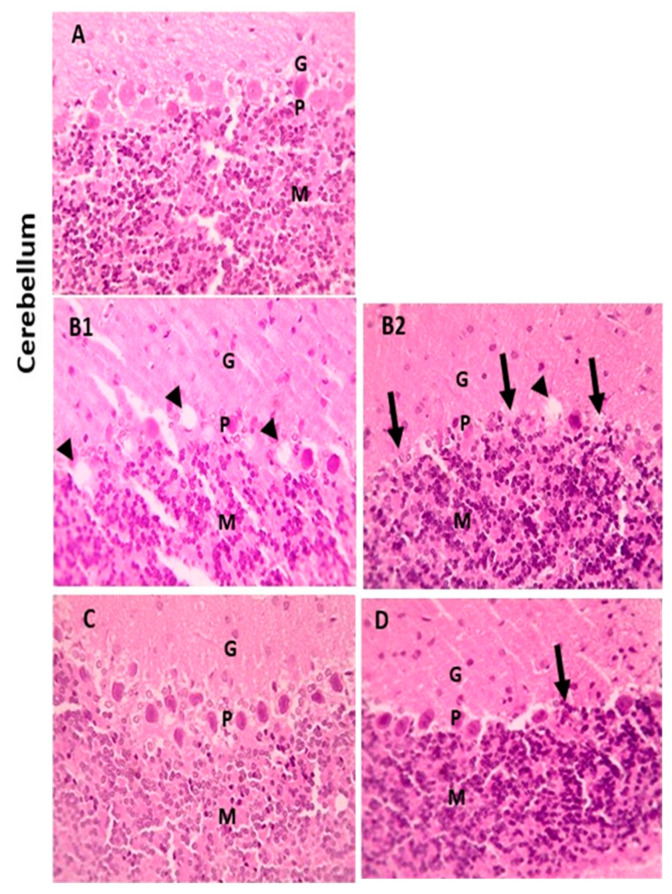
Histopathological examination of cerebellar sections using H&E. (**A**,**C**) Cerebellar section showing normal neurons in the granular (G), Purkinje (P), and molecular (M) cell layers (H&E, 400×). (**B1**,**B2**) Cerebellar section showing multiple vacuolations (black arrowheads) and prominent loss of neurons in the Purkinje layer (black arrows) (H&E, 400×). (**D**) Cerebellar section showing mild focal loss of neurons in the Purkinje layer (black arrows) (H&E, 400×).

**Figure 12 biomedicines-11-01080-f012:**
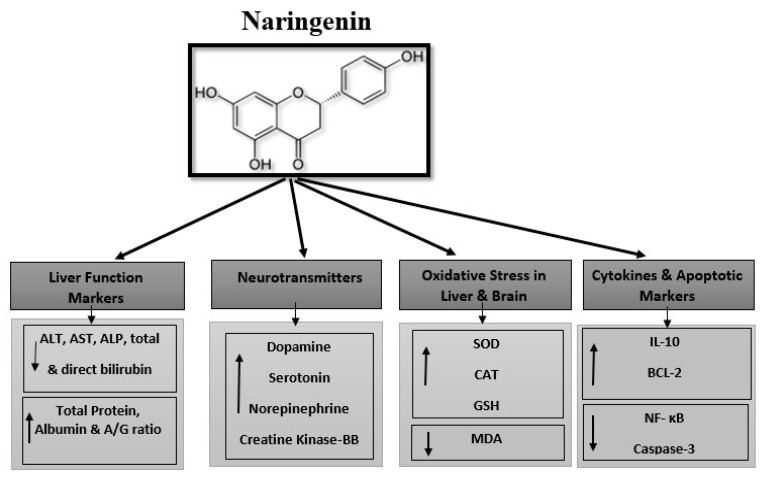
Hepato- and neuroprotective effects of naringenin against lead acetate toxicity. ↑: mean increase the value and ↓: mean decrease the value.

**Table 1 biomedicines-11-01080-t001:** Gene expression primer sequences.

Gene	Primer Sequence [5′-3′]	Reference
Rat β-actin	F:3′TCCTCCTGAGCGCAAGTACTCT5′ R: 5′GCTCAGTAACAGTCCGCCTAGAA3′	[23]
IL10	F:3′GCGGCTGAGGCGCTGTCAT5′ R: 5′CGCCTTGTAGACACCTTGGTCTTGG3′	[24]
NF-KB	F:3′GCAAACCTGGGAATACTTCATGTGACTAAG5′ R: 5′ATAGGCAAGGTCAGAATGCACCAGAAGTCC3′	[25]
BCL-2	F:3′CACCCCTGGCATCTTCTCCTT5′ R: 5′AGCGTCTTCAGAGACAGCCAG3′	[26]
Caspase-3	F:3′AGTTGGACCCACCTTGTGAG5′ R: 5′AGTCTGCAGCTCCTCCACAT3′	[27]

## Data Availability

The authors confirm that the data supporting the findings of this study are available within the article.

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
