# Peer review of "Hepatoprotective and Neuroprotective Effects of Naringenin against Lead-Induced Oxidative Stress, Inflammation, and Apoptosis in Rats"

_biomedicines, 2023, doi:10.3390/biomedicines11041080_

Round 1

Reviewer 1 Report

The manuscript documents the hepatoprotective and neuroprotective effects of naringenin on lead acetate poisoning in the rat. These results are interesting and potentially useful. However, the manuscript is very untidy and requires a thorough editorial and linguistic check and amendment.

Remarks:

Can naringenin bind lead and decrease its intake in the digestive system?

How do the applied doses of naringenin relate to its possible consumption in fruits and vegetables?

Line 22: ” /kg, BW”, rather: “/kg BW”

Line 35: “against heavy metals toxicity”, only lead toxicity was studied

Lines 45/47: Is oxidative stress the cause or effect of damage to the antioxidant system?

Lines 47/48: “”, better: “One of its negative consequences is the excessive generation of reactive oxygen and nitrogen species”

Lines 50/52: verb is lacking in the sentence

Line 73: “with protein kinase C signaling pathways, which allows it to cross the blood-brain’, unclear

Line 77: “there is available studies”, please correct grammar

Line 78: Please cite the papers

Line 94: “Ministry of Public Health”, in which respect is the Ministry invoked?

Could the author specify details of the permit for the experiment (if applicable)? 

Were the animals treated every day for 4 weeks?

Line 113: “3000 GX”, should be “3,000 × g

Line 120: please describe centrifugation conditions in g values rather than rpm

Lines 127/130: Please provide also towns of producers (distributors), not only countries

Line 133: Please describe the modification of the method

Line 139: activities rather than levels of enzymes were measured

Lines 146/147: “Spectrophotometer was used to determine how much 146

RNA was extracted”, too general, please detail the procedure

Lines 142/143: “the manufacturing technique”?

Lines 149/150: liters?

Line 153: “420 C”? “850 C”?

Lines 153/154: “It was kept at 40 C for the samples”, hard to understand

Line 158: “as a fundamental regulation”?

Lines 159-162: liters? Please give correct volumes and temperatures.

Figures: Please explain the meaning of letters (a,b,c,d)

Figure 3. Are the values expressed per mg of tissue or per mg of protein?

Line 210: “brain band isoenzyme”, please specify the isoenzyme

Page 13. Please present the graphic summary of the result as a numbered Figure.

Line 356: “BW/day/I/P”?

Line 376: “catalase and SOD degrade peroxides (OH, H2O2)”, not true. SOD does not degrade peroxides, OH is not a peroxide

Lines 392/393: “changes in cell permeability, which released cytochrome C”, please be more precise.

Author Response

Thank you for your constructive comments concerning our manuscript. We have studied your comments carefully and made major correction which we really hope to meet with your approval.

  1. Can naringenin bind lead and decrease its intake in the digestive system?

Response: Firstly, naringenin was firstly given orally for rats 2 hours before lead acetate administration to avoid any negative impacts of lead on absorption of naringenin. Additionally, pharmacokinetic analysis showed that naringenin was rapidly absorbed from the gastrointestinal tract and it was measurable in 20 min following oral administration and can reach to its beak in plasma after 2-4 hours (Kanaze et al., 2007). Naringenin administered orally during lead exposure (in the drinking water) without any evidence of affecting its absorption from GIT (Wang et al., 2012; Liu et al., 2010).

Kanaze FI, Bounartzi MI, Georgarakis M, Niopas I. Pharmacokinetics of the citrus flavanone aglycones hesperetin and naringenin after single oral administration in human subjects. European journal of clinical nutrition. 2007 Apr;61(4):472-7.

Wang J, Yang Z, Lin L, Zhao Z, Liu Z, Liu X. Protective effect of naringenin against lead-induced oxidative stress in rats. Biological trace element research. 2012 Jun;146:354-9.

Liu CM, Zheng YL, Lu J, Zhang ZF, Fan SH, Wu DM, Ma JQ. Quercetin protects rat liver against lead-induced oxidative stress and apoptosis. Environmental toxicology and pharmacology. 2010 Mar 1;29(2):158-66.

  1. How do the applied doses of naringenin relate to its possible consumption in fruits and vegetables?

Response: Doses were determined based on previous study (Uckun et al., 2020). High concentrations of naringenin are found in Grapefruits, in same time low concentrations of naringenin are also found in tomatoes and tomato-based products (Kiran et al., 2017).  Also,  Erlund et al.,  reported the blood concentration of naringenin was from <73.5 nmol/l to a mean value of 112.9 nmol/l. in healthy females dietary supplied  with  high-vegetable diet with various fruits and vegetables daily  for 2 weeks. They concluded Naringenin is bioavailable from the diet, but the plasma concentrations of naringenin are poor biomarkers of intake.

 Erlund I, M L SilasteG AlfthanM RantalaY A KesäniemiA Aro.. Plasma concentrations of the flavonoids hesperetin, naringenin and quercetin in human subjects following their habitual diets, and diets high or low in fruit and vegetables. Clinical Trial, Eur J Clin Nutr. 2002 Sep;56(9):891-8.

  1. Text (line 22):” /kg, BW”, rather: “/kg BW”

    Response: We have replaced kg, BW by kg BW throughout the manuscript.

  1. Text (line 35): “against heavy metals toxicity”, only lead toxicity was studied

Response: We have corrected this mistake.

  1. Text (Lines 45/47): Is oxidative stress the cause or effect of damage to the antioxidant system?

     Response: The mechanism of lead-induced oxidative stress involves an imbalance between generation and removal of ROS (reactive oxygen species) in tissues and cellular components causing damage to membranes, DNA and proteins (Patra et al., 2011). The effect on the antioxidant defense systems of cells is the second mechanism for lead-induced oxidative stress. Lead is shown to alter antioxidant activities by inhibiting functional SH groups in several enzymes such as ALAD, superoxide dismutase (SOD), catalase (CAT), glutathione peroxidase (GPx), and glucose-6-phosphate dehydrogenase (G6PD) (Chiba 2t al., 1996). We have corrected the sentence.

Patra RC, Rautray AK, Swarup D. Oxidative stress in lead and cadmium toxicity and its amelioration. Veterinary medicine international. 2011 Mar 20;2011.

Chiba M, Shinohara A, Matsushita K, Watanabe H, Ihaba Y. Indices of lead-exposure in blood and urine of lead-exposed workers and concentrations of major and trace elements and activities of SOD, GSH-Px and catalase in their blood. Tohoku Journal of Experimental Medicine. 1996;178(1):49–62.

  1. Text (Lines 47/48): Replace with the correct sentence.

Response: Done.

  1. Text (Lines 50/52): verb is lacking in the sentence.

Response: We have corrected the sentence.

  1. Text (Line 73): “with protein kinase C signaling pathways, which allows it to cross the blood-brain’, unclear

Response: We have corrected and cleared the sentence.

  1. Text (Line 77): “there is available studies”, please correct grammar

Response: We have corrected the sentence.

  1. Line 78: Please cite the papers

Response: We have added a reference.

  1. Text (Line 94): Could the author specify details of the permit for the experiment (if applicable)?

Response:  We have corrected the sentence and added the ethical committee.  

  1. Text (Line 113): “3000 GX”, should be “3,000 × g”

Response: We have corrected this mistake.

  1. Text (Line 120): please describe centrifugation conditions in g values rather than rpm

Response: We have corrected the value.

  1. Text (Lines 127/130): Please provide also towns of producers (distributors), not only countries

Response: We have added the value towns of producers to all used kits.

  1. Text (Line 133): Please describe the modification of the method

Response: This method is relatively easy and uses manageable sample volumes. The method has the additional advantage of saving l/3 the time, compared to the time involved to accomplish a similar analysis that was reported previously. The elimination of the alumina step results in a l/3 reduction in time to perform the analysis. The method also retains the slightly larger sample volumes, allowing the use of standard cuvettes rather than microcuvettes. Greater ease of handling the slightly larger volumes was also advantageous in this laboratory. The analysis was repeated nine more times with the result that 5-HT recovery was consistently 100% NE was in a range of 91- 100% and DA in a range of 86-93%.

  1. Text (Line 139): activities rather than levels of enzymes were measured

Response: We have corrected the sentence.

  1. Lines 146/147: “Spectrophotometer was used to determine how much 146 RNA was extracted”, too general, please detail the procedure

Response: The purity of RNA samples (1 µL) was verified by measuring their absorbance using NanoDrop spectrophotometer, ND-1000 (Thermo Scientific, USA) at 260 and 280 nm, and the RNA ratios (A260:A280) greater than 1.8 were used for further experiments.

  1. Text (Lines 142/143): “the manufacturing technique”?

Response: We have corrected the sentence.

  1. Text (Lines 149/150): liters?

Response: We have replaced l with liters throughout the manuscript.

  1. Text (Line 153): “420 C”? “850 C”?

Response: We have corrected the degree Celsius (symbol: °C) throughout the manuscript.

  1. Text (Lines 153/154): “It was kept at 40 C for the samples”, hard to understand

Response: We have corrected the degree Celsius (symbol: °C) throughout the manuscript.

  1. Text (Line 158): “as a fundamental regulation”?

Response: We have corrected the sentence.

  1. Text (Lines 159-162): liters? Please give correct volumes and temperatures.

Response: We have corrected the volumes and temperatures.

  1. Figures: Please explain the meaning of letters (a,b,c,d)

Response: It means each subscript letter that is different for each parameter in the same row of table or on different column of the same charts, shows a significant difference (P<0.05). This is meaning that a significantly different from b, c and d, and so on.  We may have a for the high value; b, c for intermediate values between a and d, while d for the lower value.

  1. Figure 3: Are the values expressed per mg of tissue or per mg of protein?

Response: According to the enclosed pamphlet, the unit of neurotransmitters was expressed as ng/mg protein. We have corrected the unit on the figure.

  1. Text (Line 210): “brain band isoenzyme”, please specify the isoenzyme

Response: CK-BB is isoenzyme of creatine kinase specified for brain tissue. We have corrected it in the figure.

  1. Page 13. Please present the graphic summary of the result as a numbered Figure.

Response: Graphical abstract summarized hepatoprotective and neuroprotective effect of naringenin against lead acetate toxicity, we have numbered the figure.

  1. Text (Line 356): “BW/day/I/P”?

Response: We have corrected the sentence.

  1. Line 376: “catalase and SOD degrade peroxides (OH, H2O2)”, not true. SOD does not degrade peroxides, OH is not a peroxide

Response: We have corrected the mistake, SOD is the first step of cellular defense to clean ROS; they convert superoxide (O2) and water (H2O) to hydrogen peroxide (H2O2) and molecular oxygen (O2). H2O2 is catalyzed by catalase into H2O and O2.

  1. Lines 392/393: “changes in cell permeability, which released cytochrome C”, please be more precise.

Response: ROS, which was predominantly produced in the mitochondria attack of membrane phospholipids and loss of mitochondrial membrane potential, which caused the intermembrane proteins, such as cytochrome c, to be released out of the mitochondria and ultimately triggered caspase-3 activation (Orrenius et al., 2007). Caspase-3 activation led to DNA breakage, nuclear chromatin condensation and cell apoptosis (Li et al., 2006)

Orrenius, S.; Gogvadze, V.; Zhivotovsky, B. Mitochondrial oxidative stress: implications for cell death. Annu. Rev. Pharmacol. Toxicol. 2007; (47): 143-183.

Li J, Tang Q, Li Y (2006). Role of oxidative stress in the apoptosis of hepatocellular carcinoma induced by combination of arsenic trioxide and ascorbic acid. Acta Pharmacol. Sin. 27: 1078-1084.

Extensive editing of English language and style required

Response: Thank you for your recommendation, our manuscript has been English edited by biomedicine editing services and we believe that now is in a good shape. (Already we have now a certificate which will be attached to the file).

Reviewer 2 Report

The manuscript entitled “Hepato-and neuro-protective effects of Naringenin on Lead toxicity induced oxidative stress, inflammation, and apoptosis markers in rats” is an interesting work but I have the following comments:

1.     In the abstract part the authors have to explain why they used Naringenin not naringin or other flavonoids like rutin in their experiment.

2.     In the abstract part, the authors have to demonsterate their results with statistical values for each separate result.

3.     In the Abstract part, the authors mentioned “In conclusion, NRG has a potential hepato-and neuro protective effect against heavy metals toxicity.”, the authors have to cplete this sentence by adding their recommendations about naringigenin.

4.     Change Lead acetate (CH3CO2)2 Pb.3H20, to be Lead acetate (CH3CO2)2 Pb.3H20 also many errors in the chemical formulas regarding numbers subscriptions.

5.     The authors mentioned in the 2.2. Experimental planning and Animals section: The Animal Research Ethical Committee of the Faculty of Veterinary Medicine at Mansoura University in Egypt developed guidelines for all animal studies, and these regulations were subscribed to throughout the whole research process” they have to add approval number to this paragraph.

6.     The discussion part needs more explanations for the MOA (mechanism of action) and SAR of Naringenin also to compare their results with other conducted experiments on other flavonoids regarding the same experiments.

7.     The conclusion part needs more figuring the outcomes and your future recommendations.

8.     Add more updated references regarding naringin like (DOI: 10.5812/jjm.65496)

9.     The whole manuscript needs language and editing polishing

Author Response

  1. In the abstract part the authors have to explain why they used Naringenin not naringin or other flavonoids like rutin in their experiment.

Response: Naringenin is one of the most important naturally-occurring flavonoid, predominantly found in some edible fruits, like Citrus species and tomatoes (Zobeiri et al., 2018). It has several biological activities such as antioxidant, antitumor, antiviral, antibacterial, anti-inflammatory, antiadipogenic and cardioprotective effects (Salehi et al., 2019). 

Zobeiri M., Belwal T., Parvizi F., Naseri R., Farzaei M.H., Nabavi S.F., Sureda A., Nabavi S.M. Naringenin and its nano-formulations for fatty liver: Cellular modes of action and clinical perspective. Curr. Pharm. Biotechnol. 2018;19:196–205.

Salehi B, Fokou PV, Sharifi-Rad M, Zucca P, Pezzani R, Martins N, Sharifi-Rad J. The therapeutic potential of naringenin: a review of clinical trials. Pharmaceuticals. 2019 Jan 10;12(1):11.

  1. In the abstract part, the authors have to demonsterate their results with statistical values for each separate result.

Response:  We have added the statistical values for each separate result.

  1. In the Abstract part, the authors mentioned “In conclusion, NRG has a potential hepato-and neuro protective effect against heavy metals toxicity.”, the authors have to cplete this sentence by adding their recommendations about naringenin.

Response:  We have added our recommendation; "However, additional research is needed in order to propose naringenin as a potential protective agent against renal and cardiac damage induced by lead acetate".

  1. Change Lead acetate (CH3CO2)2 Pb.3H20, to be Lead acetate (CH3CO2)2Pb.3H20 also many errors in the chemical formulas regarding numbers subscriptions.

Response: The correction was made.

  1. The authors mentioned in the 2.2. Experimental planning and Animals section: The Animal Research Ethical Committee of the Faculty of Veterinary Medicine at Mansoura University in Egypt developed guidelines for all animal studies, and these regulations were subscribed to throughout the whole research process” they have to add approval number to this paragraph.

Response:  Approval number has been added.

  1. The discussion part needs more explanations for the MOA (mechanism of action) and SAR of Naringenin also to compare their results with other conducted experiments on other flavonoids regarding the same experiments.

Response: We have updated our discussion section.

Similarly, it has been reported that naringenin decreases the liver marker enzymes during methyl nitrosamine-induced hepatotoxicity (Lee et al., 2004). This effect is correlated to the ability of NRG to stabilize the cell membrane in hepatic damage via its anti-lipidperoxidation activity. Another possible mechanism, by which NRG restores the hepatocellular integrity, is its ability for selective inhibition of eicosanoid synthesis, thereby reducing inflammatory responses (Mershiba et al., 2013). As well, Naringenin elicited an anti-necrotic preventive response to hepatocellular damage (Lv et al., 2013).

NRG reduced the levels of transcription factor and pro-inflammatory cytokine alleviating inflammation and cell death It has properties to produce sufficient hydroxyl substitutions, which give it the capability to scavenge ROS (Wali et al., 2020).

Furthermore, naringenin was also capable of modulating the expression of the Bcl-2 family proteins that regulate mitochondrial membrane integrity and control the release of apoptogenic factors from mitochondria.

NRG was found to inhibit acetylcholinesterase activity in the brains of rats. Accordingly, inhibition of acetylcholinesterase enzymes may perhaps contribute to the ability of NRG to attenuate memory dysfunction. Meanwhile, acetyl-cholinesterase is the enzyme responsible for metabolic degradation of acetylcholine, the major neurotransmitter implicated in learning and memory. Thus, inhibition of acetyl-cholinesterase enzymes has long been advocated as one of the major therapeutic approaches to the treatment of patients with memory deteriorations (Umukoro et al., 2018).

Lee, M. H., Yoon, S., & Moon, J. O. (2004). The flavonoid naringenin inhibits dimethylnitrosamine-induced liver damage in rats. Biological and Pharmaceutical Bulletin, 27(1), 72-76.‏

Mershiba, S. D., Dassprakash, M. V., & Saraswathy, S. D. (2013). Protective effect of naringenin on hepatic and renal dysfunction and oxidative stress in arsenic intoxicated rats. Molecular biology reports, 40(5), 3681-3691.‏

Lv, Y., Zhang, B., Xing, G., Wang, F., & Hu, Z. (2013). Protective effect of naringenin against acetaminophen-induced acute liver injury in metallothionein (MT)-null mice. Food & function, 4(2), 297-302.

Wali, B., Khattak, A. J., & Karnowski, T. (2020). The relationship between driving volatility in time to collision and crash-injury severity in a naturalistic driving environment. Analytic methods in accident research, 28, 100136

Umukoro, S., Kalejaye, H. A., Ben-Azu, B., & Ajayi, A. M. (2018). Naringenin attenuates behavioral derangements induced by social defeat stress in mice via inhibition of acetylcholinesterase activity, oxidative stress and release of pro-inflammatory cytokines. Biomedicine & Pharmacotherapy, 105, 714-723.

Prakash, O.; Singh, R.; Singh, N.; Usmani, S.; Arif, M.; Kumar, R, Ved, A. Anticancer Potential of Naringenin, Biosynthesis, Molecular Target, and Structural Perspectives. Mini Reviews in Medicinal Chemistry. 2022; 5 (22): 758-769.

  1. The conclusion part needs more figuring the outcomes and your future recommendations.

Response:  We have updated our conclusion.

  1. Add more updated references regarding naringin like (DOI: 10.5812/jjm.65496)

 Response:  We have added updated references regarding naringenin in the introduction section

 Naringenin is the aglycone form of naringin (4′,5,7- trihydroxyflavanone-7-rhamnoglucoside) which belong to the flavonoid class known as flavanones and are found mainly in citrus fruits, including lemon, orange, mandarin and grapefruit (Prakash  et al., 2022; Rivoira et al., 2021; Jaradat et al., 2018)

Prakash, O.; Singh, R.; Singh, N.; Usmani, S.; Arif, M.; Kumar, R, Ved, A. Anticancer Potential of Naringenin, Biosynthesis, Molecular Target, and Structural Perspectives. Mini Reviews in Medicinal Chemistry. 2022; 5 (22): 758-769.

Jaradat N, Shawarb N, Hussein F, Al-Masri M, Warad I, Khasati A, Shehadeh M, Qneibi M, Hussein AM, Makhamreh S. Antibacterial and antioxidant screening of semi-synthetic naringin based hydrazone and oxime derivatives. Jundishapur Journal of Microbiology. 2018 Jun 30;11(6).

Rivoira MA, Rodriguez V, Talamoni G, de Talamoni NT. New perspectives in the pharmacological potential of naringin in medicine. Current Medicinal Chemistry. 2021 Mar 1;28(10):1987-2007.

Round 2

Reviewer 1 Report

The manuscript has been considerably improved but still some details require amendment.

Line 58: please give the full name of enzyme before the acronym

Lines 59/60: please place the acronym after the enzyme full name, not at the end of the sentence

Line 92: NAR or NGR?

Line 187 and other: liters are K to avoid misunderstanding but L would be enough, according to the rule of the journal. However, the main question concerned the volumes of the reactants\; were they really in liters? If so, it would qualify for the Guinness book of records for the largest-scale PCR.

Line 433: “SOD is the first step of cellular defense to clean ROS; they convert superoxide”, please decide for either singular or plural: “SOD… it concerts”, or “SODs… they convert”

Lines 460/461: “Caspase-3 are activated DNA splitting, nuclear chromatin condensation, and cell death were all caused by Caspase-3 activation”, the sentence is completely incomprehensible. First, consider that caspases are proteases and not DNAses.

Line 492: “renewing the activities”, “renewing” is not the optimal term’ please re-phrase.

Author Response

Thank you for your constructive comments concerning our manuscript. We have studied your comments carefully and made minor correction which we really hope to meet with your approval

Response for reviewer 1 (round)2

1-Line 58: Please give the full name of enzyme before the acronym.

Response: We have added the full name in the sentence.

2-Line 59/60: Please place the acronym after the enzyme full name, not at the end of the sentence.

Response: We have corrected the place of the acronym.

3-Line 92: NAR or NRG?

Response: It was mistyping, we have corrected it to NRG.

4-Line 187 and other: liters are K to avoid misunderstanding but L would be enough, according to the rule of the journal. However, the main question concerned the volume of the reactants\; were they really in liters? If so, it would qualify for the Guinness book of records for the largest-scale PCR.

Response: many thanks for your accurate observation, It was mistyping, we have corrected it to μl.  

5-Line 433: "SOD is the first step of cellular defense to clean ROS; they convert superoxide", please decide for either singular or plural: "SOD… it concerts", or "SODs… they convert"

Response: It was mistyping, we have corrected the sentence.

6-Line 460/461: "Caspase-3 are activated DNA splitting, nuclear chromatin condensation, and cell death were all caused by Caspase-3 activation", the sentence is completely incomprehensible. First, consider that caspases are proteases and not DNAses.

Response: It was mistyping, we have corrected the sentence.

7-Line 492: "renewing the activities", "renewing" is not the optimal term' please re-phrase.

Response: we have changed it to restoring.

Reviewer 2 Report

The authors conducted all the required corrections and I have not any more suggestions. 

Author Response

The authors conducted all the required corrections and I have not any more suggestions. 

Response: thank you very much for your recommendation